# A Comparative Study of Immunogenicity, Antibody Persistence, and Safety of Three Different COVID-19 Boosters between Individuals with Comorbidities and the Normal Population

**DOI:** 10.3390/vaccines11081376

**Published:** 2023-08-17

**Authors:** Fatemeh Ashrafian, Fahimeh Bagheri Amiri, Anahita Bavand, Mahsan Zali, Mona Sadat Larijani, Amitis Ramezani

**Affiliations:** 1Clinical Research Department, Pasteur Institute of Iran, Tehran 1316943551, Iran; fatemeh.ashrafian24@gmail.com (F.A.); anahita.bavand@gmail.com (A.B.); mahsan.zali@gmail.com (M.Z.); 2Department of Epidemiology and Biostatistics, Research Centre for Emerging and Reemerging Infectious Diseases, Pasteur Institute of Iran, Tehran 1316943551, Iran; drop.dr.bagheri@gmail.com

**Keywords:** COVID-19 vaccine, booster, underlying diseases, antibody persistence

## Abstract

Data on immunogenicity, immune response persistency, and safety of COVID-19 boosters in patients with comorbidities are limited. Therefore, we aimed to evaluate three different boosters’ immunogenicity and safety in individuals with at least one underlying disease (UD) (obesity, hypertension, and diabetes mellitus) with healthy ones (HC) who were primed with two doses of the BBIBP-CorV vaccine and received a booster shot of the same priming vaccine or protein subunit vaccines, PastoCovac Plus or PastoCovac. One hundred and forty subjects including sixty-three ones with a comorbidity and seventy-seven healthy ones were enrolled. The presence of SARS-CoV-2 antibodies was assessed before the booster injection and 28, 60, 90, and 180 days after it. Moreover, the adverse events (AEs) were recorded on days 7 and 21 postbooster shot for evaluating safety outcomes. Significantly increased titers of antispike, antiRBD, and neutralizing antibodies were observed in both UD and HC groups 28 days after the booster dose. Nevertheless, the titer of antispike IgG and anti-RBD IgG was lower in the UD group compared to the HC group. The long-term assessment regarding persistence of humoral immune responses showed that the induced antibodies were detectable up to 180 days postbooster shots though with a declined titer in both groups with no significant differences (*p* > 0.05). Furthermore, no significant difference in antibody levels was observed between each UD subgroup and the HC group, except for neutralizing antibodies in the hypertension subgroup. PastoCovac Plus and PastoCovac boosters induced a higher fold rise in antibodies in UD individuals than BBIBP-CorV booster recipients. No serious AEs after the booster injection were recorded. The overall incidence of AEs after the booster injection was higher in the UD group than the HC group among whom the highest systemic rate of AEs was seen in the BBIBP-CorV booster recipients. In conclusion, administration of COVID-19 boosters could similarly induce robust and persistent humoral immune responses in individuals with or without UD primarily vaccinated with two doses of the BBIBP-CorV. Protein-based boosters with higher a higher fold rise in antibodies and lower AEs in individuals with comorbidities might be considered a better choice for these individuals.

## 1. Introduction

As of 20 January 2023, 663,640,386 confirmed cases of COVID-19 and 6,713,093 deaths were globally reported by the WHO [1]; among which 7,562,998 confirmed cases of COVID-19 and 144,728 deaths occurred in Iran [2]. On 11 May 2023, the WHO declared the end of coronavirus disease 2019 (COVID-19) as a global public health emergency [3]. However, monitoring the impact of COVID-19 and the effectiveness of prevention and control strategies remains a public health priority [4]. Various types of studies have demonstrated that people with comorbidities are at a higher mortality risk when they develop COVID-19 [5,6], which could be a major concern for these individuals and also an important challenge for the global health system. Considering the potential threat of COVID-19 morbidity and severity in this population, prevention of infection/reinfection is a crucial priority. There is no doubt that vaccination is one of the most successful and cost-effective ways to control the SARS-CoV-2 spread [7]. Although some COVID-19 vaccines have been widely approved [8], the safety and immunogenicity of COVID-19 vaccines in specific groups such as those with chronic diseases have not been fully understood. Some data have revealed lower immunogenicity of COVID-19 vaccines in populations with obesity [9,10,11,12], diabetes [13,14,15,16], and hypertension [9,17,18] compared to healthy subjects. Hence, it seems that there is an association between comorbidities and lower antibody levels, especially regarding inactivated COVID-19 vaccines. Moreover, the antibody persistency of COVID-19 boosters in these populations has not been determined, yet. Therefore, a booster dose might be essential in these people in order to boost the immune response safely.

From another point of view, some studies have shown that protein subunit vaccines could be administrated as a better alternative booster following two doses of inactivated vaccine [19,20,21]. Two protein subunit COVID-19 vaccines, PastoCovac (Soberana 02) and PastoCovac Plus (Soberana Plus) have been manufactured at the Pasteur Institute of Iran in collaboration with the Finlay Vaccine Institute in Cuba. Although recent studies have confirmed the safety and immunogenicity of both vaccines in healthy adults [22,23,24] and children [25,26], there are not efficient associated data of these vaccines among populations with comorbidities. Therefore, we retrospectively evaluated the immunogenicity and safety of three COVID-19 boosters in people with underlying diseases in comparison with healthy individuals who had received two doses of the BBIBP-CorV vaccine and received a booster shot of the same priming vaccine or protein subunit vaccines, PastoCovac Plus or PastoCovac. The comparative assessment was conducted to address any potential differences between three booster types in healthy ones and those with comorbidities. The present data could provide a proper booster choice for cases with obesity, hypertension and diabetes mellitus. Moreover, the durability of humoral immune response after a COVID-19 booster vaccination in specific individuals was evaluated and compared to the healthy ones.

## 2. Materials and Methods

### 2.1. Study Design and Participants

We conducted a retrospective study to evaluate the safety, immunogenicity, and antibody persistency of different COVID-19 boosters in individuals with underlying diseases in comparison with healthy subjects after primary vaccination with the BBIBP-CorV.

This study was conducted on 140 participants who were referred to Pasteur Institute of Iran between February 2022 and August 2022 to be vaccinated and subsequently agreed to be followed in a 180-day follow-up schedule. Data on the demographic and clinical features of all participants were collected through face-to-face or phone interviews. 

The participants aged ≥18 years who were primed with BBIP-CorV (Sinopharm, Beijing CNBG with an interval of 4–5 weeks) were investigated as below:(1)BBIP-CorV primed/PastoCovac Plus boosted (BP);(2)BBIP-CorV primed/PastoCovac boosted (BPa);(3)BBIP-CorV primed/BBIP-CorV boosted (BB).

The booster shot was administrated 3–6 months post the second dose in all groups.

Participants were divided into two UD or HC groups according to their recorded medical history and profiles. UD group was defined as people with at least one of the three conditions including obesity, hypertension, and diabetes mellitus. Therefore, the UD group was divided into three subgroups including obesity, hypertension, and multi.

People with a body mass index (BMI) ≥ 30 kg/m^2^ have been considered obese people. Hypertension and diabetes mellitus were diagnosed by physicians’ screen based on a medical history and relevant medicine therapy. According to CDC guidelines, people with diabetes mellitus are diagnosed with high level of fasting blood sugar (≥126 mg/dL) [27]. Moreover, hypertensive patients were defined as having high blood pressure (with systolic blood pressure ≥ 130 mmHg and diastolic blood pressure ≥ 80 mmHg) [28]. People with more than one mentioned disease are categorized in multi subgroup. In addition, those with normal BMI and without any underlying diseases were considered the HC group.

Exclusion criteria were other comorbidities, pregnant women, and vaccinated participants who had been primed with other COVID-19 vaccines rather than BBIP-CorV or did not meet the standard interval time.

All study participants who met criteria of the study and agreed to participate were provided a consent form. The study was approved by the Ethics Committee of the Pasteur Institute Iran (reference number: IR.NREC.1400.020) and all the applied methods were in accordance with the Declaration of Helsinki.

### 2.2. Antibodies Response

Blood sera samples were collected before booster administration and 28, 60, 90, and 180 days after it. The levels of antispike IgG antibody, antiRBD IgG, and neutralizing antibody were measured using Anti-SARS-CoV-2 Quantivac ELISA (IgG) (Euroimmun, Lübeck, Germany), antiRBD IgG (Idealtashkhis, Tehran, Iran), and SARS-CoV-2 Neutralizing Ab (Pishtazteb, Iran), respectively. Thresholds of ≥2.5, 5, and 11 IU/mL were considered positive for SARS-CoV-2 Neutralizing Ab, antiRBD SARS-CoV-2, and antispike SARS-CoV-2 IgG, respectively. The antibody concentration above the threshold was repeated by diluting the samples with diluent solution.

### 2.3. Safety Assessment

All participants were monitored for adverse reactions (AEs) for 30 min after receiving the booster shot. For evaluating late adverse events, all the local (pain at the injection site, soreness, induration, swelling, and warmness) and systemic adverse events (fever, chill, weakness, headache, fatigue, nausea, vomiting, diarrhea, and arthralgia) were recorded in the questionnaire via phone calls on days 7 and 21 after the booster dose. The classification of adverse events was performed based on the related guidelines.

### 2.4. Statistical Analysis

Descriptive analysis were reported as geometric mean and 95% CI, median, and interquartile range (IQR) for quantitative variables and frequency and percent for qualitative variables. Normality of quantitative variables were checked using Kolmogorov–Smirnov test. For each antibody, fold rise was calculated by dividing the titer of the antibody on day 28, 60, 90, and 180 by day zero. The distribution of both raw and fold rise of each antibody was assessed between the two groups using the Mann–Whitney U test. In addition, antibody titers amount during time points (0, 28, 60, 90 and 180) were assessed using Friedman test.

For both groups (UD and HC) separately, the difference in antibody fold rise among the different subgroups in each time point was calculated using the Mann–Whitney U test and Kruskal–Wallis test. All the tests were considered two-way and a *p*-value < 0.05 was reported as statistically significant. All statistical analyses and graphs were performed using SPSS statistics software (version 26.0) and GraphPad Prism (version 9.0.0).

## 3. Results

### 3.1. Study Population

This study recruited 140 participants including 77 HC and 63 UD. The demographics are shown in Table 1. All participants were divided into age subgroups ≤ 40 years old and >40 years old.

Of all participants with UD, 57.14% and 25.40% had obesity and hypertension and 17.46% had more than one comorbidity. Patients with diabetes mellitus were included in the multi subgroup due to the coexistence of their disease with other comorbidities. 

The COVID-19 history showed that 14.3% of the UD group and 23.4% of the HC ones had a history of COVID-19. In the UD group, 41.3%, 27%, and 31.7% of patients received PastoCovac Plus, PastoCovac, and BBIBP-CorV as booster doses, respectively. In the HC group, 28, 27, and 22 participants belonged to BP, BPa, and BB groups, respectively.

### 3.2. Comparison of COVID 19 Booster Simmunogenicity and Antibody Persistency between UD and HC Groups

To determine whether the underlying disease is associated with humoral immune responses, we examined postbooster antibody levels in UD and HC participants. In addition, the baseline of GMT values regarding antispike IgG, antiRBD IgG, and neutralizing antibodies were compared to the antibody levels on days 28, 60, 90, and 180 to assess booster vaccines for both groups.

The titers of anti-spike IgG, antiRBD IgG, and neutralizing antibodies were not significantly different in both UD and HC groups over time (Table 2). The assessment of antibody titer showed that both healthy individuals and the UD group reached a significant titer rise 28 days after the booster dose (Figure 1B–D). Nevertheless, the titer rise of antispike IgG and antiRBD IgG was lower in the UD group compared to the HC group. A decline in the level of all antibody titers was observed on days 60, 90, and 180 compared to day 28 (Table 2, Figure 1B–D). However, this trend was not significantly different between healthy and UD groups. Antibody persistence results indicated that neutralizing antibodies were the most persistent during 180 days postbooster shots (Figure 1B–D).

The titers and fold rise of specific antibodies between the HC and UD at different time points are shown in Table 2. Furthermore, the titer of antibodies of each UD subgroup were compared with the HC group and the results showed no significant difference in antibody levels (Figure 2A–F,H,I). Nevertheless, neutralizing antibody levels on day 28 was significantly lower in the hypertension subgroup (*p* = 0.000) compared to the HC group (Figure 2G).

#### 3.2.1. Immunogenicity and Antibody Persistency of Boosters among People with Underlying Diseases

The possible association between the fold rise of specific antibodies and demographic parameters such as age, sex, as well as COVID-19 history, types of UD, and boosters were also investigated.

The median fold rise of antispike IgG on day 28 was higher in UD participants aged >40 years (7.69, IQR: 14.26) compared to younger participants (1.61, IQR: 6.07) (*p* = 0.02). The fold rise of antiRBD IgG on day 60 in men (9.70, IQR: 46.30) was significantly higher than in women (*p* = 0.03). Also, the fold rise distribution of neutralizing antibody on day 60 was significantly different between females (1.05, IQR: 0.13) and males (1.23, IQR: 1.39) (*p* = 0.03).

Fold rise of neutralizing antibody on day 28 was significantly different among sub-groups of UD (*p* = 0.04), obesity (1.09, IQR: 0.39), hypertension (1.6, IQR: 2.03), and multigroup (1.5, IQR: 8.13).

A significant difference in the antispike IgG fold rise on day 28 was observed among the booster groups of UD participants (*p* < 0.001); the BB group (1.55, IQR: 1.19) showed a lower fold rise of antispike IgG in comparison to both BP (11.01, IQR:36.75) and BPa (11.34, IQR: 20.25) groups. Furthermore, the fold rise of neutralizing antibody on day 28 showed a significant difference between UD participants receiving PastoCovac Plus (1.66, IQR: 3.04) compared to BBIBP-CorV booster (1.03, IQR: 0.14).

In the UD group, a high rate of antiRBD IgG fold rise was observed in the BP (15.26, IQR: 121.92) and BPa (13.85, IQR: 23.06) groups than the BB group (15.26, IQR: 121.92) on days 28 (*p* < 0.001, *p* = 0.03), while the PastoCovac Plus injection had the greatest effects. Moreover, similar results were observed on day 90, with a greater increase in the antiRBD IgG fold rise in the BP group (11.57, IQR: 151.55) than in the BB group (11.57, IQR: 151.55).

On day 60, there was a significant difference in antiRBD IgG fold rise among three booster groups (*p* = 0.03), while BPa (9.38, IQR: 45.09) showed the highest fold rise in comparison to other indicated groups ([BB: 1.05, IQR: 0.94], [BP: 4.74, IQR: 20.33]) (Table 2). Other variables had no significant impact on the antibodies’ median fold rise on 28, 60, 90, and 180 days following the booster shot (Table 3).

#### 3.2.2. Immunogenicity and Antibody Persistency of Boosters among Healthy People

The fold rise of antispike IgG on day 180 in healthy individuals aged >40 years (3.14, IQR: 10.32) was significantly higher than in subjects aged ≤40 (0.64, IQR: 2.41) (*p* = 0.04). In the HC group, the fold rise of neutralizing antibody on day 60 was significantly different among the booster groups, the BP group (1.51, IQR: 5.11) showed higher neutralizing antibody fold rise compared to other boosters (*p* = 0.04). On day 28, there was a significant difference in antiRBD IgG fold rise among booster groups, while the BP (6.81, IQR: 155.93) showed the highest fold rise compared to both BPa (2.02, IQR: 11.16) and BB (1.49, IQR: 4.67) groups (*p* = 0.02). Moreover, a lower rate of antiRBD IgG fold rise was observed in healthy people with a COVID-19 history on days 28 (*p* = 0.02), 60 (*p* = 0.01), and 90 (*p* = 0.02). In healthy subjects, other variables revealed insignificant effects on the fold rise of antibodies after booster administration (Appendix A).

### 3.3. Comparative Safety Outcomes of COVID-19 Boosters between UD and HC Groups

To investigate whether comorbidities were associated with the incidence of adverse events, we assessed the adverse events incidence after booster administration in participants with UD. No serious AEs were observed within 30 min after the booster injection. Fourteen people (eight healthy people and six UD people, *p* < 0.99) reported local AEs within seven days after receiving a booster dose. Fifteen out of all participants reported at least one systemic AE. Furthermore, systematic AEs were significantly higher in the UD group (17.5%) in comparison with healthy people (5.2%, (*p* = 0.01). The most frequently reported local AE in the UD and HC groups was pain at the injection site (9.5%, 10.4%) (Figure 3A). Weakness was the most common systemic AE and which reported significantly more in the UD group (*p* = 0.01) (Figure 3B). On day 21 after booster vaccination, six people reported local AEs, with injection-site pain being the most common AE. Moreover, seven people reported at least one systemic AE, of which weakness was the most common. There was higher local and systemic AEs incidence within day 21 in the HC group compared to the UD group. However, the difference between the two groups was not statistically significant.

Among booster subtypes, more local AEs (13.6%) were observed in the BPa group compared with the BP (5.6%) and BB (11.9%) groups, but it was not significant (*p* = 0.37). BBIBP-CorV injected people reported significantly more systemic AE (21.4%) than the BP (1.9%) and BPa (9.1%) groups (*p* = 0.005).

Based on subgroup analysis, there were no reported differences in local AEs between the UD and HC groups, while those with UD who had injected BBIBP-CorV reported more systemic AEs (40%) than other booster groups (*p* = 0.005) (Figure 3C,D).

## 4. Discussion

Based on the available scientific evidence, people with underlying diseases (e.g., obesity, hypertension, etc.) are at risk of increased rate of severe complications and mortality induced by COVID-19 due to an impaired and dysregulated immune response [5,6]. Therefore, vaccination is a top priority in these populations [7]. Inactivated vaccines are the most widely administered vaccine type in Iran, especially among the elderly. Many published results have reported a decline in antibodies titer after primary vaccination with the BBIBP-CorV vaccine and the positive effects of boosters to restore diminish antibodies [29,30]. Therefore, the identification of booster type that can induce stronger immunogenicity in BBIBP-CorV immunized UD people is important, as the data on this issue is limited. In the current study, for the first time, we assessed the presence of SARS-CoV-2 antibodies before and after booster vaccination during 6 months of follow-up in individuals with underlying diseases. Moreover, the safety, immunogenicity, and antibody persistency of protein-based and inactivated booster vaccines were compared in specific populations.

Our results revealed that antispike, antiRBD, and neutralizing antibody titers significantly increased on day 28 after the booster dose. This assessment indicates the impact of protein subunit vaccines on BBIBP-CorV primed individuals. Consistent with our study, no significant difference in antibody titers was observed between Chinese participants with and without underlying medical conditions after vaccination with Vero cell-derived inactivated COVID-19 vaccine [31]. Moreover, a previous study comparing the immunogenicity of CoronaVac vaccine between people with comorbidities and healthy controls showed that there was no significant difference in neutralizing antibody levels between both groups at different time points [32]. Another study confirmed similar immunogenicity of inactivated COVID-19 vaccine between elderly subjects with hypertension or/and diabetes mellitus and the healthy group [33]. Conversely, some investigations reported lower antibody titers in patients with comorbidities postvaccination in comparison with healthy people [13,14,15,16,34]. Although we observed that antibody titers had declined by days 60, 90, and 180, they remained at an acceptable level over 180 days among which neutralizing antibodies were the most persistent ones. In the same manner, two investigations confirmed the persistence of antibody titers several months after vaccination in hypertensive patients [17,35]. Moreover, the persistence of neutralizing antibody of the PastoCovac Plus booster was reported in Cubans eight months after primary vaccination with the PastoCovac vaccine [36]. These findings have noted the positive effects of booster vaccines on production and persistency of antibody response in immunized people living with UD.

In our study, we evaluated the association between the fold rise of specific antibodies and demographic parameters such as age and sex, as well as types of UD and boosters. We found that the antibodies’ fold rise was not different between age subgroups. However, the antispike IgG fold rise on day 28 was different in the UD group aged >40 than younger people, which could be due to a higher baseline antibody level in this subgroup or/and a higher number of participants of this category compared to healthy people. In agreement with our results, a study revealed that the two doses of CoronaVac inactivated vaccines induced higher neutralizing antibody levels in senior people and cancer patients than in younger and healthy people [32]. In contrast, some studies reported lower SARS-CoV-2 antibody levels in older people [37,38,39,40]. Therefore, previous studies and the present results demonstrate antibody response to SARS-CoV-2 after a booster dose in people with older age enhanced, emphasizing the excellent immunogenicity of the booster dose in the population with comorbidities and older age.

Various types of studies established that patients with comorbidities showed different postvaccination humoral immune responses [31,35,41]. We found that people with hypertension had lower levels of neutralizing antibodies on day 28 compared to the HC group. In agreement with our results, two recent studies revealed that antispike and antiRBD antibodies after primary vaccination with the CoronaVac vaccine were significantly lower in the hypertension group than in the healthy group [17,35]. Therefore, designing and implementing the vaccination program with more efficiency in this subgroup seems necessary.

Several studies have shown that men experience more severe COVID-19 outcomes [42,43,44] and sex differences affect the immune responses to COVID-19 vaccines [45]; though, the exact cause of this gender-based difference is not clear. We found that antibody titers did not discriminate by sex, except for antiRBD and a fold rise in neutralizing antibodies on day 60. Similarly, a meta-analysis study showed greater efficacy of the COVID-19 vaccines in men than in women [46]. One study showed that antibody response to a COVID-19 vaccine at one and five months after primary vaccination was not affected by sex, while antispike IgG levels differed between women and men at three months postvaccination [17]. In addition, another study has reported no significant difference in antibody titers between men and women after the booster dose [47]. Differences regarding the impact of sex on antibody levels may be due to the coexistence of other host factors such as age and other comorbidities, and further clinical trials are necessary to determine and evaluate the impact of sex differences on the effectiveness of the COVID-19 vaccine.

Good immunogenicity and safety of protein subunit COVID-19 vaccines have been reported in the recent data [48,49]. We demonstrated that the fold rise in antibodies in different window times was significantly different among booster recipients. The lowest fold rise in antibodies was observed in UD people after receiving the BBIBP-CorV booster. In agreement with our findings, the higher rate of neutralizing antibody induced by NVSI-06-07 booster (protein-based vaccine) was detected in comparison with the BBIBP-CorV homologous regimen [50]. Our previous study also showed the better boosting effect of PastoCovac and PasctoCovac plus as a booster vaccine on humoral immune response in comparison to the inactivated vaccine in Iranian adults [51]. Moreover, a recent study reported that both inactive and recombinant SARS-CoV-2 protein boosters had the same immunogenicity and were safe and tolerable in cancer patients and healthy subjects [52]. Overall, our research has revealed that the protein subunit vaccines appear to be able to induce a stronger humoral immune response than the inactivated COVID-19 vaccine.

Besides immunogenicity, we surveyed the safety outcome of inactivated and protein-based COVID-19 boosters in people with UD. In our study, no serious AEs were found after the booster injection. We observed that the overall incidence of adverse events within 7 days was higher in the UD group than in the HC group. The most common local AE in healthy and UD people was pain at the injection site. Weakness is the most common systemic AE in people with UD. Among booster subgroups, the systemic AEs were significantly higher in UD individuals who received the BBIBP-CorV booster than in other groups. These results indicated that protein subunit vaccines could be proposed as a better alternative to booster doses with lower AEs in people with UD. Similarly, a study performed by Li et al. demonstrated that the incidence of injection-site pain, fatigue, and headache was higher in some disease groups than in healthy control after primary vaccination with CoronaVac vaccine [53]. Moreover, no AEs were observed in people with obesity and hypertension following an inactivated COVID-19 vaccination [31]. Overall, booster vaccines, especially protein-based boosters, have great immunogenicity, excellent safety, and therefore could be recommended for people with underlying diseases.

There are some limitations in the current study. According to the type of study, retrospective, the population study was small. Second, cellular immunity was not evaluated, which could possibly be performed in further studies to determine whether participants with a low rate of antibody response could develop adequate T-cell response to prevent COVID-19. Further long-term studies are required to evaluate the persistence of cellular immune responses in participants with chronic diseases after a booster vaccination. This may be important in determining the schedule of subsequent vaccinations.

What this study adds to the previous data:The humoral immune responses in people with obesity, hypertension, and diabetes mellitus were induced after three kinds of COVID-19 booster vaccines.There was no significant difference between the healthy individuals and those with underlying diseases regarding humoral immune responses.The protein subunit vaccine had a better impact on immune response induction in people with comorbidities.The induced antibodies were detectable 180 days after the booster shots in both healthy and comorbidity groups.Less adverse events were recorded after PastoCovac/Plus booster shots compared to BBIBP-CorV.

## 5. Conclusions

In conclusion, our data provide the first comprehensive picture of the safety, immunogenicity, and antibody persistency of COVID-19 three booster types in Iranian cases with comorbidities in comparison with healthy ones. Different types of booster vaccines could elicit strong and persistent humoral immune responses in BBIBP-CorV primed individuals in both healthy and comorbidity groups. Among UD people, protein-based booster vaccines had a higher fold rise in antibodies and lower AEs in comparison with inactivated booster vaccines. Owing to the fact that UD subjects are a high-risk group, PastoCovac/Plus protein subunit vaccines are recommended with an excellent immunogenicity and safety profile.

## Figures and Tables

**Figure 1 vaccines-11-01376-f001:**
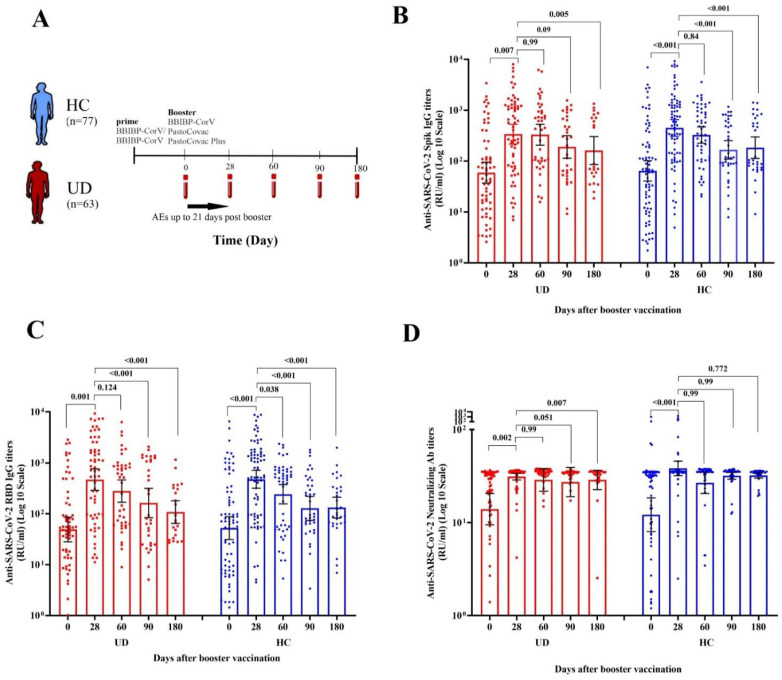
Comparative time-course analysis of antibody response to COVID-19 vaccination. (**A**) COVID-19 vaccination study design. (**B**) The titers of antispike IgG antibodies, (**C**) antiRBD IgG, and (**D**) neutralizing antibodies in the healthy control and people with underlying diseases over time. HC: healthy control, UD: participants who had at least one comorbidity. *p* value < 0.05 was considered significant.

**Figure 2 vaccines-11-01376-f002:**
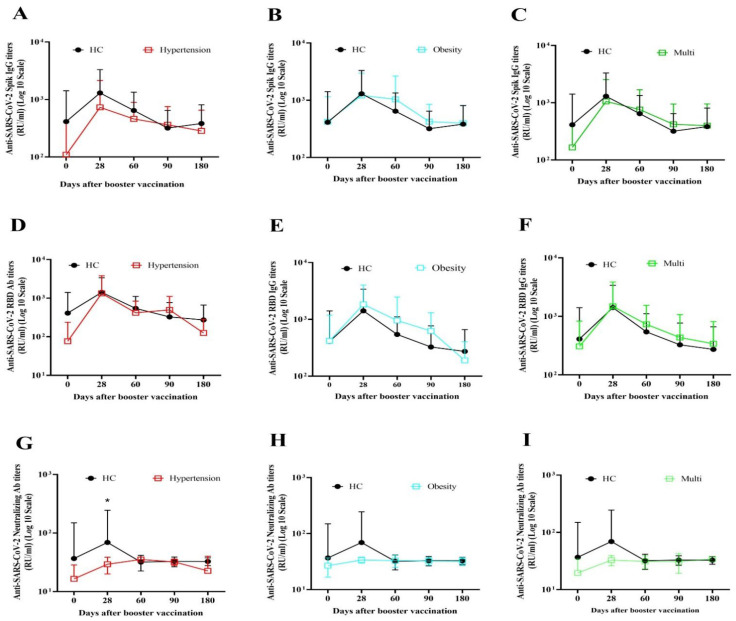
Comparative analysis of responses of specific antibodies to COVID-19 vaccination stratified by the presence of underlying diseases. The antibody titers before vaccination (baseline) up to 180-day follow-up demonstrated in participants with different comorbidities in comparison to the healthy control group (HC). Anti-SARS-CoV-2 Spike IgG titers in (**A**) hypertension, (**B**) obesity, and (**C**) multi subgroups. Anti-SARS-CoV-2 RBD IgG titers in (**D**) hypertension, (**E**) obesity, and (**F**) multi subgroups. Anti-SARS-CoV-2 neutralizing antibody titers in (**G**) hypertension, (**H**) obesity, and (**I**) multi subgroups. * *p* < 0.05 was considered statistically significant. The Mann–Whitney U test was used to compare titers of each antibody at each time between HC and UD groups.

**Figure 3 vaccines-11-01376-f003:**
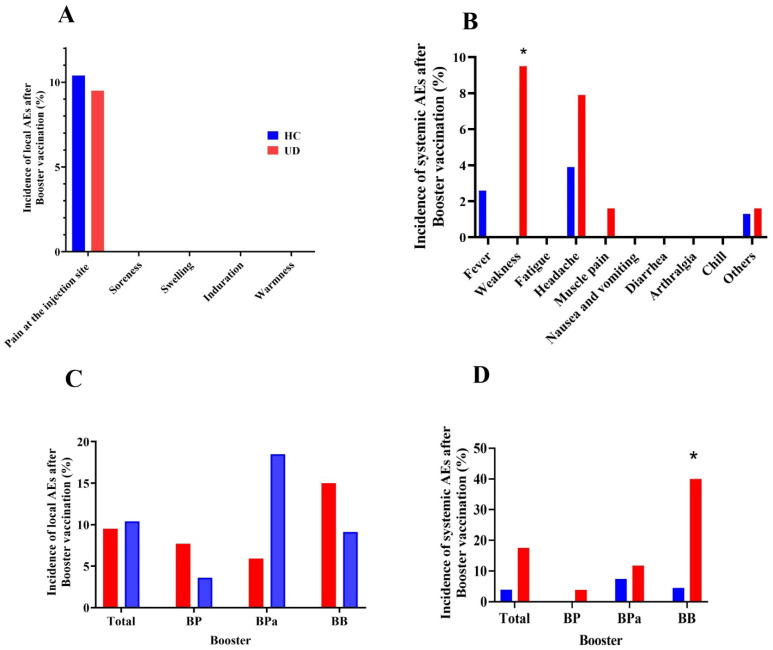
Incidence of adverse events reported within 7 days after booster vaccination. Incidence of (**A**) local and (**B**) systemic AEs in healthy and UD participants. Incidence of (**C**) local and (**D**) systemic AEs between the HC and UD groups base on booster groups. HC: healthy control, UD: participants who had at least one comorbidity. * *p* < 0.05 was considered statistically significant.

**Table 1 vaccines-11-01376-t001:** The baseline clinical characteristics of the study participants.

	Total N (%)	HC N (%)	UD N (%)
Gender			
Female	74 (52.9%)	41 (53.2%)	33 (52.4%)
Male	66 (47.1%)	36(46.8%)	30 (47.6%)
Age			
≤40	57 (40.7%)	38 (49.4%)	19 (30.2%)
>40	83 (59.3%)	39 (50.6%)	44 (69.8%)
COVID-19 History			
No	113 (80.7)	59 (76.9)	54 (85.7)
Yes	27 (19.3)	18 (23.4)	9 (14.3)
Booster			
BP	54 (38.6%)	28 (36.4%)	26 (41.3%)
BPa	44 (31.4%)	27 (35.1%)	17 (27.0)
BB	42 (30.0)	22 (28.6%)	20 (31.7%)

HC: healthy control, UD: participants who had at least one comorbidity (obesity, hypertension, and diabetes mellitus). BP: BBIBP-CorV/PastoCovac-Plus, BPa: BBIBP-CorV/PastoCovac, BB: BBIBP-CorV/BBIBP-CorV.

**Table 2 vaccines-11-01376-t002:** Anti-SARS-CoV-2 Spike, anti-SARS-CoV-2 RBD, and neutralizing antibody levels and fold rise between healthy and UD participants in different time points.

	HC	UD	*p* Value
Anti-SARS-CoV-2 Spike Antibody (Geometric Mean, 95% CI)			
Antibody titers			
Day 0	64.26 (40.53−101.88)	59.22 (36.87−95.12)	0.86
Day 28	449.52 (301.75−656.59)	339.09 (216.83−530.28)	0.38
Day 60	325.87 (224.88−472.21)	330.00 (205.34−530.34)	0.97
Day 90	166.11 (109.63−251.69)	189.99 (113.92−316.86)	0.79
Day 180	183.41 (112.74−298.35)	161.22 (85.65−303.49)	0.39
Mean Fold rise			
Day 28	7.00 (5.01−9.76)	5.72 (3.92−8.37)	0.31
Day 60	3.88 (2.33−6.46)	4.27 (2.35−7.77)	0.75
Day 90	1.67 (0.95−2.94)	2.88 (1.29−6.45)	0.25
Day 180	1.54 (0.69−3.41)	1.61 (0.56−4.65)	0.98
Neutralizing Antibody (Geometric Mean, 95% CI)			
Antibody titers			
Day 0	12.16 (8.01−18.47)	13.97 (9.47−20.62)	0.78
Day 28	38.31 (31.95−45.95)	31.27 (28.79−33.96)	0.054
Day 60	26.84 (20.61−34.96)	28.85 (21.82−38.15)	0.90
Day 90	31.91 (29.31−34.74)	27.32 (19.04−39.22)	0.10
Day 180	32.22 (30.14−34.45)	28.79 (22.66−36.57)	0.34
Mean Fold rise			
Day 28	9.14 (5.98−13.96)	9.52 (5.24−19.30)	0.55
Day 60	3.51 (2.01−6.12)	4.10 (2.04−8.23)	0.95
Day 90	1.78 (1.22−2.60)	1.55 (0.87−2.78)	0.32
Day 180	1.52 (1.17−1.97)	1.66 (0.80−3.45)	0.83
Anti-SARS-CoV-2 RBD Antibody (Geometric Mean, 95% CI)			
Antibody titers			
Day 0	52.38 (31.24−87.83)	49.12 (28.07−85.95)	0.88
Day 28	478.74 (319.48−717.38)	470.89 (290.65−762.90)	0.98
Day 60	242.97 (156.99−376.03)	279.99 (169.23−463.24)	0.34
Day 90	128.02 (74.30−220.57)	163.21 (84.18−316.43)	0.77
Day 180	132.60 (82.72−212.55)	107.96 (64.67−180.24)	0.34
Mean Fold rise			
Day 28	3.15 (2.13−4.66)	2.23 (1.52−3.27)	0.27
Day 60	1.61 (1.03−2.49)	1.78 (1.19−2.66)	0.54
Day 90	1.31 (0.66−2.61)	3.70 (1.27−10.78)	0.39
Day 180	1.05 (0.44−2.54)	1.29 (0.37−4.45)	0.11

Fold rise was calculated using dividing the titer of the antibody on days 28, 60, 90, and 180 by day zero. HC: healthy control, UD: participants who had at least one comorbidity (obesity, hypertension, and diabetes mellitus). The distribution of GMT was assessed using the Mann–Whitney U test at each daytime point.

**Table 3 vaccines-11-01376-t003:** Median and IQR of anti-SARS-CoV-2 Spike, anti-SARS-CoV-2 RBD, and neutralizing antibody fold rise on days 28, 60, 90, and 180 in people with underlying disease.

	Day 28		Day 60		Day 90		Day 180	
	Median (IQR)	P	Median (IQR)	P	Median (IQR)	P	Median (IQR)	P
**Anti-SARS-CoV-2 Spike Antibody Median Fold Rise (IQR)**								
Age								
≤40	1.61 (6.07)	**0.02**	2.73 (9.32)	0.55	1.37 (4.77)	0.22	0.92 (3.30)	0.36
>40	7.69 (14.26)		4.12 (25.18)		3.90 (14.19)		2.77 (17.39)	
Gender								
Female	3.45 (11.14)	0.3	1.90 (5.53)	0.08	1.33 (11.02)	0.13	0.86 (9.89)	0.33
Male	5.80 (18.23)		6.84 (30.94)		5.02 (11.19)		3.49 (14.48)	
COVID-19 History								
No	4.58 (12.96)	0.62	4.04 (17.82)	0.19	3.86 (11.04)	0.63	1.86 (14.40)	0.66
Yes	4.30 (11.63)		1.14 (25.51)		2.37 (12.60)		0.75 (18.53)	
Types of UD								
Hypertension	9.39 (34.58)	0.21	4.12 (19.25)	0.91	5.26 (86)	0.54	19.53 ()	0.25
Obesity	3.92 (10.58)		2.45 (25.18)		3.86 (12.44)		0.92 (14.43)	
Multi	4.48 (14.03)		3.56 (6.34)		1.55 (4.73)		0.98 (2.28)	
Booster								
BP	11.01 (36.75)	**<0.001**	6.07 (31.60)	0.33	5.62 (110.79)	0.53	1.43 (64.41)	0.83
BPa	11.34 (20.25)		6.05 (28.82)		4.42 (11.83)		3.50 (14.96)	
BB	1.55 (1.19)		2.18 (2.53)		1.45 (2.86)		1.14 (1.95)	
**Anti-SARS-CoV-2 Neutralizing Antibody Median Fold Rise (IQR)**								
Age								
≤40	1.05 (0.52)	0.19	1.05 (0.43)	0.28	1 (0.46)	0.69	0.96 (0.16)	0.36
>40	1.18 (1.81)		1.11 (0.85)		1.12 (0.69)		1.03 (0.67)	
Gender								
Female	1.09 (1.16)	0.4	1.05 (0.13)	**0.03**	1 (0.19)	0.17	0.99 (0.20)	0.92
Male	1.14 (1.54)		1.23 (1.39)		1.46 (0.68)		1.01 (11.86)	
COVID-19 History								
No	1.12 (1.38)	0.11	1.10 (0.96)	0.61	1.10 (0.71)	0.3	0.98 (0.58)	0.84
Yes	1.02 (0.15)		1.08 (0.16)		0.99 (0.15)		1.10 (0.23)	
Types of UD								
Hypertension	1.60 (2.03)	**0.04**	1.18 (1.03)	0.57	1.57 (170.96)	0.73	0.96 ()	0.94
Obesity	1.09 (0.39)		1.08 (0.19)		1.10 (0.54)		1.01 (0.37)	
Multi	1.50 (8.13)		1.13 (4.80)		0.99 (2.77)		1.01 (11.47)	
Booster								
BP	1.61 (3.04)	**0.01**	1.05 (1.18)	0.1	1.50 (2.19)	0.29	1.03 (60.69)	0.42
BPa	1.11 (0.61)		1.20 (0.90)		1.11 (0.63)		1.01 (0.59)	
BB	1.03 (0.14)		1.03 (0.18)		1 (0.10)		0.96 (0.10)	
**Anti-SARS-CoV-2 RBD Antibody Median Fold Rise (IQR)**								
Age								
≤40	2.28(32.18)	0.73	1.36(11.32)	0.35	0.35(9.53)	0.05	0.35(0.17)	0.12
>40	5.10(29.58)		4.22(18.20)		8.73(20.61)		1.18(11.28)	
Gender								
Female	2.29 (30.07)	0.07	1.28 (4.23)	**0.02**	0.82 (19.25)	0.24	0.039 (2.54)	0.47
Male	10.75 (36.62)		9.70 (46.30)		9.39 (21.23)		0.72 (20.92)	
COVID-19 History								
No	4.69 (34.15)	0.78	3.19 (17.45)	0.35	1.43 (22.66)	0.53	0.52 (4.61)	0.96
Yes	6.38 (27.27)		0.98 (15.66)		1.10 (19.35)		0.66 (5.26)	
Types of UD								
Hypertension	6.38 (38.46)	0.66	2.17 (21.07)	0.92	11.56 (192.97)	0.32	37.53 ()	0.18
Obesity	5.10 (29.81)		1.80 (17.13)		1.62 (19.23)		0.40 (2.99)	
Multi	4.63 (17.70)		1.95 (9.35)		0.65 (13.25)		0.38 (2.58)	
Booster								
BP	15.26 (121.92)	**<0.001**	4.74 (20.33)	**0.03**	**11.57 (151.55)**	0.03	0.54 (34.19)	0.72
BPa	13.85 (23.06)		9.83 (45.09)		8.73 (19.05)		0.95 (5.66)	
BB	1.26 (0.57)		1.05 (0.94)		**0.34 (0.67)**		0.39 (0.55)	

UD: participants who had at least one comorbidity (obesity, hypertension, and diabetes mellitus). BP: BBIBP-CorV/PastoCovac-Plus, BPa: BBIBP-CorV/PastoCovac, BB: BBIBP-CorV/BBIBP-CorV. Bold P values are shown as significant.

## Data Availability

The data that support the findings of this study are available from the corresponding author upon reasonable request.

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
