# Peer review of "A Comparative Study of Immunogenicity, Antibody Persistence, and Safety of Three Different COVID-19 Boosters between Individuals with Comorbidities and the Normal Population"

_vaccines, 2023, doi:10.3390/vaccines11081376_

Round 1

Reviewer 1 Report

This manuscript describes the significant findings and contribution to the field of Covid 19 vaccines. However, following some major concerns are identified that require the authors’ attention:-

1.      Was there any significant difference in the humoral responses among the homologous and heterologous booster groups in healthy controls? The author should provide the data for the same.

2.      Different studies have demonstrated that SARS-CoV-2-specific T cell responses are essential for viral clearance, may prevent infection without seroconversion. The authors should have assessed the T cell responses in the different groups.

3.      Numbering is wrong in the reference section. Please corrected from reference number 3 onwards.

4.      Line 461: Please correct the typo error “CIVID” as “COVID”.

5.      Please correct the year as “2023” in the header section.

6.      The authors have stated that Protein-based heterologous dose is considered a better choice particularly for individuals with co-morbidities.

But, previous prime boost studies  have shown that mRNA vaccines, viral vectored vaccines and inactivated vaccines against COVID 19, used for heterologous booster vaccination  has generated better response than the homologous vaccination. (Costa Clemens et al, Lancet. 2022 Feb 5;399 (10324):521-529; Robert et al, N Engl J Med 2022; 386:1046-1057). The authors are required to justify the claim.

English language used in fine though there are some typographical errors.

Author Response

Dear Editor, 

On behalf of all the authors, I would like to thank you and the reviewers for their careful review of our manuscript. The suggestions were incorporated into the text of the manuscript and all the changes are marked in blue. It should be noted that the reference list and page and line numbers are now updated due to amending the suggested changes by the reviewers.

Reviewer 1

  1. Was there any significant difference in the humoral responses among the homologous and heterologous booster groups in healthy controls? The author should provide the data for the same.
  • Thanks for your worthy recommendation. We analyzed and included the difference humoral responses among the homologous and heterologous booster groups in healthy controls in the text and the table.

  1. Different studies have demonstrated that SARS-CoV-2-specific T cell responses are essential for viral clearance, may prevent infection without seroconversion. The authors should have assessed the T cell responses in the different groups.
  • Thank you for your valuable suggestion. This comment is right, however, the evaluation of cellular response was not the aim of this study according to the limited funding. This issue has been mentioned as a limitation at the end of the discussion part.
  1. Numbering is wrong in the reference section. Please corrected from reference number 3 onwards.
  • The order has been corrected.
  1. Line 461: Please correct the typo error “CIVID” as “COVID”.
  • The error was corrected.
  1. Please correct the year as “2023” in the header section.
  • 2021 in the footer and header has been created by the journal, not the authors.
  1. The authors have stated that Protein-based heterologous dose is considered a better choice particularly for individuals with co-morbidities.

But, previous prime-boost studies have shown that mRNA vaccines, viral vector vaccines, and inactivated vaccines against COVID-19, used for heterologous booster vaccination have generated better responses than the homologous vaccination. (Costa Clemens et al, Lancet. 2022 Feb 5;399 (10324):521-529; Robert et al, N Engl J Med 2022; 386:1046-1057). The authors are required to justify the claim.

  • In Clemens et al. study, the boost effects of the third heterologous dose were of either a recombinant adenoviral vectored vaccine (Ad26.COV2-S, Janssen), an mRNA vaccine (BNT162b2, Pfizer–BioNTech), or a recombinant adenoviral-vectored ChAdOx1 nCoV-19 vaccine (AZD1222, AstraZeneca), compared with a third homologous dose of CoronaVac in healthy adults. The study showed that all four vaccines administered as a third dose induced a significant increase in binding and neutralizing antibodies, which could improve protection against infection. Heterologous boosting resulted in more robust immune responses than homologous boosting and might enhance protection. Meanwhile, in our study, we evaluated and compared three different boosters (PastoCovac Plus and PastoCovac (two protein-based boosters), BBIBP-CorV (inactivated booster)) in individuals who received two doses of the BBIBP-CorV vaccine including underlying diseases and healthy cases as control. Our results demonstrated that administration of COVID-19 boosters can equally induce robust and persistent humoral immune responses in individuals with or without UD primarily vaccinated with 2-doses of the BBIBP-CorV. Protein-based boosters with higher antibodies' fold rise and lower AEs in individuals with comorbidities might be considered a better choice for these individuals.

Although both studies evaluated the effects of different boosters on the humoral immune response of participants who received two doses of inactivated COVID-19 vaccine, the immunogenicity of protein-based boosters, as well as the immunogenicity of boosters on individuals with underlying diseases in addition to the healthy group, was not evaluated in the Clemens study.

  1. English language used in fine though there are some typographical errors.
  • The text has been rechecked thoroughly.

Reviewer 2 Report

After careful evaluation, the submission needs to meet the standard requirements for publication. I have compiled a comprehensive review highlighting the major issues that need to be addressed before the manuscript can be considered for further review.

Title and Abstract:

The title needs more clarity to convey the central focus of the study. Additionally, the abstract lacks essential information such as the research question, methodology, and key findings. I recommend revising both the title and abstract to reflect the scope and importance of the research accurately.

Introduction:

The introduction needs to be more developed to provide an overview of the existing key literature. The research gap and rationale for the study are unclear. Please provide a clear articulation of the need for this research and how it contributes to the existing knowledge in the field.

Methodology:

The methodology section lacks the necessary details to allow for reproducibility. Key information such as sample size, data collection procedures, and statistical analyses are either missing or inadequately described. It is crucial to provide a thorough and transparent presentation of the research methodology.

Results and Discussion:

The results and discussion sections need to be more adequately presented. The results should be organized and accompanied by appropriate statistical analyses. Additionally, the discussion should interpret the results in light of the research question and relevant literature, drawing meaningful conclusions and providing insights into the implications of the findings.

References:

The reference list is incomplete and lacks recent and pertinent sources. Please ensure that all references cited in the manuscript are included and properly formatted according to the journal's guidelines.

In conclusion, while the topic of your research is potentially of interest to our readership, the manuscript requires substantial revisions before it can be reviewed for scientific quality fairly.

Language and Presentation:

The manuscript requires significant improvement in terms of language and writing style. The text contains grammatical errors, and unclear sentences. I strongly recommend editing to enhance the clarity and readability of the manuscript.

Author Response

Dear Editor, 

On behalf of all the authors, I would like to thank you and the reviewers for their careful review of our manuscript. The suggestions were incorporated into the text of the manuscript and all the changes are marked in blue. It should be noted that the reference list and page and line numbers are now updated due to amending the suggested changes by the reviewers.

Title and Abstract:

The title needs more clarity to convey the central focus of the study. Additionally, the abstract lacks essential information such as the research question, methodology, and key findings. I recommend revising both the title and abstract to reflect the scope and importance of the research accurately.

  • Thank you for your careful recommendation. The title and abstract have been improved.

Introduction:

The introduction needs to be more developed to provide an overview of the existing key literature. The research gap and rationale for the study are unclear. Please provide a clear articulation of the need for this research and how it contributes to the existing knowledge in the field.

  • Thank you for the recommendation. PastoCovac/Plus are two protein subunit vaccines that have been manufactured in Pasteur Institute of Iran in collaboration with Cuba. The data regarding protein vaccines against COVID-19 are limited as they have not been widely applied worldwide. The clinical trials and the follow-up studies have been conducted in Iran as well as Cuba. The long-term follow-up studies on COVID-19 vaccination are of a high priority as they were developed so fast due to the pandemic imposed challenges. The related studies have been published and cited through the text (the other ones are under review). Among these data, the population with comorbidities was neglected which led to the present study. The research gap has been included in lines 50-66 in addition to the abstract.

Methodology:

The methodology section lacks the necessary details to allow for reproducibility. Key information such as sample size, data collection procedures, and statistical analyses are either missing or inadequately described. It is crucial to provide a thorough and transparent presentation of the research methodology.

  • Thank you for the comment. According to the concept of the study in the retrospective format, we analyzed the data of the available population (of long-term previous study DOI: 1093/femspd/ftad010) who met the criteria of the study after informed consent reception.  Therefore, it was not possible to make a sample size. The data collection has been fully described in section 2.1.

            We completed the statistical analysis as below:

"Descriptive analysis was reported as geometric mean and 95%CI, median and interquartile range (IQR) for quantitative variables and frequency and percent for qualitative variables. The normality of quantitative variables was checked using the Kolmogorov-Smirnov test. For each antibody, fold rise was calculated using dividing the titer of the antibody on day 28, 60, 90, and 180 by day zero. The distribution of both raw and fold rise of each antibody was assessed between the two groups using the Mann-Whitney U test. In addition, antibody titers amount during time points (0, 28, 60, 90, and 180) were assessed using the Friedman test.

For both groups (UD and HC) separately, the difference in antibody fold rise among the different subgroups, at each time point was calculated using the Mann-Whitney U test and Kruskal-Wallis test. All the tests were considered two-way and a p-value < 0.05 was reported as statistically significant. All statistical analyses and graphs were performed using SPSS statistics software (version 26.0) and GraphPad Prism (version 9.0.0)." 

Results and Discussion:

The results and discussion sections need to be more adequately presented. The results should be organized and accompanied by appropriate statistical analyses. Additionally, the discussion should interpret the results in light of the research question and relevant literature, drawing meaningful conclusions and providing insights into the implications of the findings.

  • Thank you for the comment. Based on the goal of each research question, we analyzed the antibody titers as quantitative variables. Because the distribution of data was not normal, so we used non-parametric tests. For comparison of the distribution of antibody titer or fold-rise of them, we used the Mann-Whitney U test (for comparison 2 group, table 2 and 3 and Figure 2) or the Kruskal-Wallis test (for more than 2 groups, Table 3 and supplementary table 1). As the response of antibody titers in each time-point was important, the amount of repeated measures of each antibody was calculated by the Freedman test. We tried to select the best analyzes based on the objectives, however, we welcome and appreciate your suggestion for any additional analysis that would help to provide better results in line with the objectives. The discussion part has been also improved.

References:

The reference list is incomplete and lacks recent and pertinent sources. Please ensure that all references cited in the manuscript are included and properly formatted according to the journal's guidelines.

  • The reference list is completed and formatted according to the journal's guidelines.

The manuscript requires significant improvement in terms of language and writing style. The text contains grammatical errors and unclear sentences. I strongly recommend editing to enhance the clarity and readability of the manuscript.

  • Thank you for your recommendation; all grammatical errors and unclear sentences were corrected throughout the manuscript.

Reviewer 3 Report

The manuscript by Ashrafian et al., describes the immune responses and side effects following booster vaccination with different types of vaccines against SARS-Cov-2.

The manuscript is well-written with good English and scientific soundness. Although the study is limited to regions that use these specific vaccines, it still provides an insight to the efficacy of mixed vaccination regimens. 

good quality, well written

Author Response

Dear Editor,

On behalf of all the authors, I would like to thank you and the reviewers for their careful review of our manuscript. The suggestions were incorporated into the text of the manuscript and all the changes are marked in blue. It should be noted that the reference list and page and line numbers are now updated due to amending the suggested changes by the reviewers.

Reviewer 3

The manuscript is well-written with good English and scientific soundness. Although the study is limited to regions that use these specific vaccines, it still provides an insight to the efficacy of mixed vaccination regimens. 

We do appreciate your great attention. According to other reviewers’ comments, all changes were highlighted in the manuscript.

Round 2

Reviewer 2 Report

The authors presented a submission entitled "Immunogenicity, Antibody Persistence, and Safety of COVID-19 Boosters in Individuals with Comorbidities and Normal Populations (Iran)." I find the study's topic and design to be valuable; however, after getting the details on statistics, I have a major concern that needs to be addressed before the paper can be considered for revision.

One crucial aspect that caught my attention is the absence of power analysis in the study. A power analysis is an essential step in experimental design as it helps determine the appropriate sample size needed to detect statistically significant effects or differences between study groups. Without a power analysis, it becomes challenging to evaluate the reliability and validity of your findings.

This analysis should clearly specify the effect sizes you expected to observe, the desired level of statistical power (e.g., 80% or higher), the chosen significance level (typically set at 0.05), and any assumptions made about the data distribution. Justifying your selected effect sizes with references or previous studies would also strengthen your analysis.

By including the power analysis, the readers will better understand the robustness of your findings and interpretations.

I appreciate the importance of your research in shedding light on the immunogenicity, antibody persistence, and safety of COVID-19 boosters in different populations. Addressing the concern raised regarding the power analysis will significantly improve the quality of your submission and ease our review process.

Minor upgrades will improve the take home message

Author Response

Dear Editor-In-Chief

We do appreciate the great attention to our data. As it has been described in the manuscript, the data distribution was not normal in the study according to the retrospective design. Therefore, nonparametric tests were applied to compare the outcomes between the groups and subgroups. As far as we are concerned, routine tests are not applied to calculate the power of nonparametric analysis, and the calculated powers are based on the mean and SD.

Nevertheless, in this study non- parametric assessment is based on median and IQR not the mean.

 Although the power of small sample size studies is low, such studies can generally help to provide evidence in the future.

Thank you in advance and hope the response is convincing.